# Case Report of Myelodysplastic Syndrome in a Sickle-Cell Disease Patient Treated with Hydroxyurea and Literature Review

**DOI:** 10.3390/biomedicines10123201

**Published:** 2022-12-09

**Authors:** Pagona Flevari, Ersi Voskaridou, Frédéric Galactéros, Giovanna Cannas, Gylna Loko, Laure Joseph, Pablo Bartolucci, Justine Gellen-Dautremer, Emmanuelle Bernit, Corine Charneau, Anoosha Habibi

**Affiliations:** 1Centre of Excellence in Rare Hematological Disease-Hemoglobinopathies, Laiko General Hospital, 11527 Athens, Greece; 2Sickle Cell Referral Center, Department of Internal Medicine, Henri-Mondor University Hospital, APHP, U-PEC, 94000 Créteil, France; 3Hospices Civils de Lyon, Edouard-Herriot Hospital, Internal Medicine, Reference Centre for Sickle-Cell Disease, Thalassemia and Other Red Blood Cell Disorders, 69003 Lyon, France; 4Martinique Hospital, 97212 Martinique, France; 5Biotherapy Department, Necker Children’s Hospital, Assistance Publique-Hôpitaux de Paris, 75610 Paris, France; 6Antilles-Guyane Reference Centre for Sickle-Cell Disease, Thalassemia and Other Red Blood Cell Disorders, Pointe à Pitre, 97157 Guadeloupe, France

**Keywords:** case report, hydroxyurea, myelodysplastic syndrome, sickle cell disease

## Abstract

The safety profile of hydroxyurea (HU) in patients with sickle-cell disease (SCD) is relatively well known. However, despite the suspected association of HU with myeloid neoplasms in myeloproliferative neoplasms (MPN), and the publication of sporadic reports of myeloid malignancies in SCD patients treated with HU, the possible excess risk imparted by HU in this population having an increasing life expectancy has failed to be demonstrated. Herein, we report one case of myelodysplastic syndrome emanating from the results on safety and effectiveness of HU on the largest European cohort of 1903 HU-treated adults and children who were followed-up prospectively in an observational setting over 10 years, accounting for a total exposure of 7309.5 patient-years. A comparison of this single case with previously published similar cases did not allow us to draw any significant conclusions due to the paucity of these events.

## 1. Introduction

Sickle-cell disease (SCD) refers to a group of inherited hemoglobin disorders characterized by sickling of erythrocytes when they are deoxygenated, leading to a significant burden of acute and chronic complications. Patients with SCD have acute and chronic pain due to vaso-occlusive pain crisis (VOCs) and have a reduced quality of life. The presence of hemoglobin A (HbA) and/or hemoglobin F (HbF) decreases the sickling and decelerates the polymerization process. Since the 1990s, hydroxyurea (HU) has been a cornerstone for the prevention of vaso-occlusive crises, the most frequent complication in SCD patients. HU is a chemotherapeutic agent that has been used for decades for the treatment of myeloproliferative neoplasms (MPNs). As a ribonucleotide reductase inhibitor, it interferes with DNA synthesis and repair mechanisms, causing cell cycle arrest and allowing γ globin genes to be more actively expressed. HU alters the kinetics of erythroid proliferation by enhancing the production of more F cells from primitive progenitors and subsequently the production of HbF. An increased risk of developing malignancies in SCD patients, especially myeloid neoplasms, emerged from epidemiological studies with incidence rates between 0.09 and 0.18% [1]. Despite the suspected association of HU with myeloid neoplasms, and the publication of sporadic reports of myeloid malignancies in SCD patients treated with HU, the possible excess risk imparted by HU in this population having an increasing life expectancy has failed to be demonstrated. Suspected mechanisms are chronic inflammation and disease-related immunomodulation, with potential additional risk of genomic alterations.

In the framework of the prospective observational study ESCORT-HU (NCT02516579), which evaluated the long-term safety and effectiveness of HU tablets in SCD patients across several European centers, one incident hematological malignancy was reported and described there [2]. Similarly, a combined analysis of all published MDS cases found in the scientific literature is also provided.

## 2. Case Report

The patient was a 40-year-old male with homozygous HbSS SCD enrolled in the ESCORT-HU study. Because of recurrent VOC episodes, he started HU treatment at the age of 23. At the time of inclusion, at the age of 30, he had experienced seven episodes of painful crises over the previous year (lasting for more than 48 h) and was switched to HU tablets at a dose of 25 mg/kg/day. He had no other complications related to SCD aside from priapism. Further improvement of the hematological and clinical response was shown by an increase in fetal hemoglobin rising from 20 to 26% and the absence of painful episodes during the first three-year follow-up in the study. The subsequent clinical course was notable for one episode of acute chest syndrome and painful crisis requiring hospitalization and the initiation of exchange transfusion at the age of 37. Because of iron overload, deferasirox was prescribed at the age of 38 and was maintained for only 20 days due to the subsequent clinical course. Mild heart failure (left ventricular ejection fraction of 55%) was detected at the age of 38 and was related to the underlying SCD. Severe pulmonary arterial hypertension (grade 4–5) was confirmed by right heart catheterization at the age of 41 during a further hospitalization due to an infection. A few months later, HU was withdrawn (after 17 years of exposure) because of a decreased hemoglobin level ranging from 8.7 to 7.0 g/dL and a decrease in the platelet count ranging from 160 to 116 × 10^9^/L. Three months later, the patient was hospitalized for fever and the peripheral blood count at that time revealed these information: red blood cells: 2.55 × 10^12^; reticulocyte count: 8.59%; hemoglobin: 7.4 g/dL; leucocyte count: 24.11 × 10^9^/L; and platelet count: 45 × 10^9^/L. Examination of the bone marrow revealed hyperplasia and dyserythropoiesis of the red blood cell lineage, left-shift granulopoiesis, and also dysmegakaryopoiesis with an elevated number of megakaryocytes. There were 6% CD34^+^ blasts of the granulocyte lineage and also intermediate bone marrow infiltration (12–15%) by CD3^+^, CD2^+^, CD8+, CD5^+^, CD56^+^ (partially), CD7^-^, CD57^-^, TdT^-^, TIA-1-T lymphocytes. Thus, the diagnosis of refractory anemia with an excess of blasts-1 (RAEB-1) was confirmed. Cytogenetic analysis demonstrated a complex clonal abnormality including deletions of 5 q, 3 p, and 7 p, and monosomies of chromosomes 16, 17, and 18. There was no transformation into acute myeloid leukemia (AML). While the patient was screened for hematopoietic stem-cell transplantation (HSCT), the severe and persistent cytopenia led to his death three months after the established diagnosis.

## 3. Discussion

Conventionally, myeloid neoplasia occurring after exposure to a cytotoxic drug is classified as therapy-related. The WHO classification considers a single entity for t-AML (therapy-induced acute myeloid leukemia) and t-MDS (therapy-related myelodysplastic syndrome), in contrast to their de novo counterparts. The t-MDS/t-AML morphological classification has presumably limited relevance in predicting outcome, whereas cytogenetic stratification appears to provide more relevant information in terms of latency and prognosis [3].

A more frequent pattern of genetic rearrangements relative to their de novo counterparts has been identified for certain cytotoxic compounds such as alkylating agents/radiation therapy involving chromosomes 5 [del(5 q)] and/or 7 [-7/del(7 q)] and a complex karyotype, with a poor prognosis (median survival time of eight months). This pattern was also associated with a long latency (5–7 years) and with MDS, with a rapid progression into AML. A strong correlation has also been found between cytogenetic rearrangements leading to 17 p deletion and p53 mutations [4], which is the most encountered mutation in t-MDS/t-AML that occurs at a 2- to 4-fold higher frequency when compared to de novo myeloid neoplasms [5,6]. The complex chromosomal rearrangement exhibited by our patient had probably a detrimental impact on the MDS outcome. However, the cytogenetic pattern that was observed did not clearly superimpose with the abnormalities induced by alkylating agents.

Although the pathogenic mechanism is still unclear, mounting evidence indicates that instead of directly inducing mutations, chemotherapy (when combined with high dose radiations in some patients) may favor clonal expansion of selected progenitor or stem cells harboring mutations in specific genes (such as p53) by suppressing competing hematopoiesis—leading then to transformation into MDS/AML after subsequent acquisition of driver mutations [7].

The common observation of specific karyotypic abnormalities following the use of HU in MPN raised suspicion about the potential leukemogenicity of HU in this population [8], in addition to the intrinsic propensity of MPN to transform into MDS/AML even without prior cytotoxic therapy. However, case control and prospective studies with extended follow-up were unable to detect a clear excess risk that could be distinguished from concomitant cytotoxic exposure and the natural evolution of more progressive MPN.

In SCD patients, one British study suggested an increased risk of hematological malignancies, especially AML. A chronic inflammatory state, free radicals induced by free hemoglobin or heme, chronic hematopoietic stress, and premature “aging” of hematopoiesis with accrued genetic, epigenetic, and functional alteration of hematopoietic cells were suggested as possible explanations for this finding. HU may carry itself an intrinsic risk of oncogenic potential due to the increased frequency of acquired DNA mutations in SCD patients [9]. Nevertheless, no significant increase in myeloid malignancies occurred in a large population-based cohort following introduction of HU therapy in SCD patients. The largest cohort study in SCD patients treated with HU with a follow-up of up to 17 years did not provide evidence for a specific risk for patients treated with HU at conventional doses [10].

We retrieved further reports of MDS from the Medline database via Pubmed until May 2020, using keywords such as “sickle-cell disease”, “sickle-cell anemia”, “myelodysplastic” and “leukemia”. These cases are summarized in Table 1 and cumulative analysis of these reports is provided below. We chose to restrict the search to MDS to obtain further insights into the pathogenesis of the preleukemic process without the involvement of additional abnormalities acquired during transformation into AML. In total, five additional cases have been reported [11,12,13,14,15]. Importantly, even with a low estimation of 10% of patients treated with HU (from among 22,000 SCD patients in Europe and approximately 90,000 in the US), this reporting rate appears to be low.

All patients were male and relatively young as the median age at diagnosis was 42 years, with two notable outliers: one 55-year-old patient without prior exposure to HU (case 2), and one 34-year-old (case 4) with a limited exposure of two years having received non-myeloablative chemotherapy and radiotherapy for HSCT with subsequent graft loss. All other patients were treated with HU for a median duration of 14 years (range: 10–20).

The long latencies of onset after starting HU (median: 15 years) that were observed for the five patients treated with HU at conventional doses were longer than those observed in t-MDS/AML patients treated with cytotoxic agents or immunosuppressants in non-malignant conditions (median 10.8 years). Abnormalities in chromosomes 5, 7, or both, in 4 of the 5 cases (case 1, 3–5) involving HU exposure were also found, consistent with the known cytogenetic profile of therapy-related myelogenous neoplasm observed in MDS, as well as in the case without prior exposure to HU (case 2). This latter patient, who was also older than the other patients of this series, showed a deletion in the short arm of chromosome 17, thus suggesting a common pathogenic pathway with the leukemogenicity of alkylating agents. In contrast, in the younger patient in case 4, who had a limited exposure to HU (only 2 years), the disease displayed shorter latency with partial loss of the long arm of chromosome 7 (but no reported alteration of chromosome 17). This patient had also received a non-myeloablative regimen and radiation, and the seven-year latency was more consistent with the commonly reported timelines associated with t-MDS.

Our patient developed MDS at an age comparable to the ones reported in other cases following a relatively similar duration of exposure to HU. Karyotype analysis of blast cells demonstrated features more commonly reported for patients with t-MDS, but no clear-cut cytogenetic profile has emerged when compared to the cases without prior exposure to HU.

Despite the abovementioned concerns, HU has a beneficial effect in many chronic complications of SCD and this is demonstrated in Figure 1. The death rate in patients treated with HU was significantly lower than that of SCD patients treated conventionally and without HU (13 deaths in HU group vs. 49 deaths in non-HU group). Figure 2 demonstrates that the probability of 10-year survival was 86% for the patients treated with HU (HU patients) vs. only 65% for the non-HU patients.

Additionally, HU is no longer the only available drug to relieve SCD patients from VOCs, improving their quality of life and increasing their life span. Crizanlizumab and Voxelotor are new emerging agents that are added to the management of SCD with HU, which still remains the gold standard.

## 4. Conclusions

One case of MDS in a 40-year-old male with homozygous HbSS SCD was collected in the prospective ESCORT-HU cohort study among the 1906 participants, of whom 55% were adults (total exposure of 7309.5 patient years). No other hematological malignancy was collected. Thus, the incidence of hematological malignancy appears to be low. Aside from the long duration of treatment with HU for SCD, no specific risk factor was identified for our patient, but additional surveillance with a longer follow-up is required.

## Figures and Tables

**Figure 1 biomedicines-10-03201-f001:**
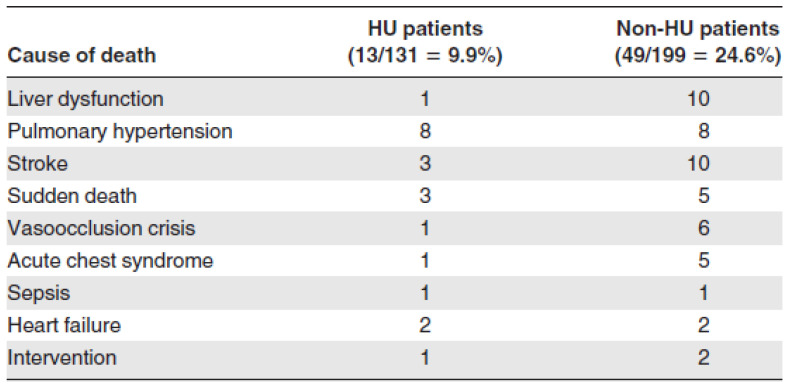
Causes of death in HU and non-HU patients (adapted from [10]).

**Figure 2 biomedicines-10-03201-f002:**
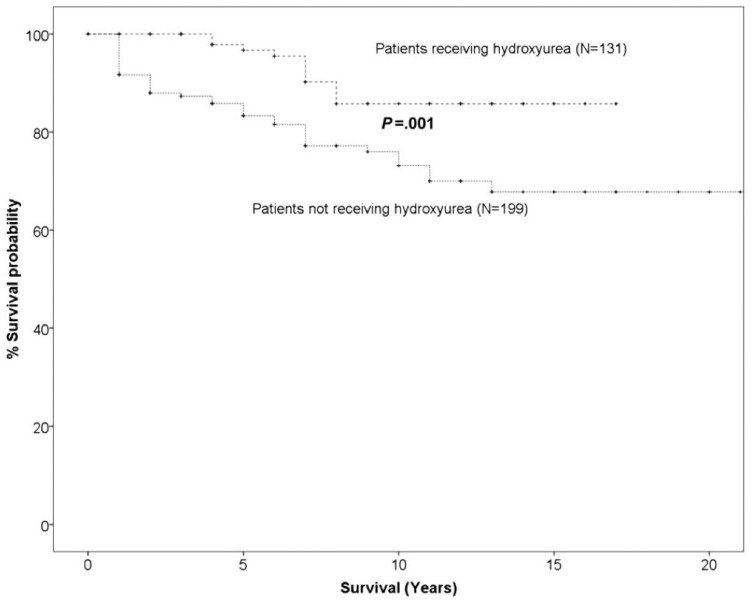
Probability of 10-year overall survival in HU SCD patients versus non-HU SCD patients (adapted from [10]).

**Table 1 biomedicines-10-03201-t001:** Pathological features of MDS (past and present case)–only MDS according to the WHO 2016 classification were included (i.e., not including cases of MDS/MPN) ^&^.

Case Nr, Ref, Country	Case 1, US [11]	Case 2, UK [12]	Case 3, FR [13]	Case 4, US [14]	Case 5, GR (Current Report)	Case 6, US [15]
**Gender**	Male	Male	Male	Male	Male	Male
**Hb beta-chain genotype**	Not specified	S/S	S/S	S/β^0^	S/S	S/S
**Treatment (daily dose)**	Transfusions, HU (1.5 to 2.0 g)	none	Transfusions, HU (1 to 1.5 g)	Transfusion, HU (0.4 to 1.5 g) then matched-sibling donor HSCT with unspecified non-myelo-ablative therapy and radiation, then supportive therapy	Regular transfusions HU	HU then autologous HSCT with lentiviral vector encoding anti-sickling beta-globin, with busulfan myeloablative conditioning
**Cumulative HU exposure**	15 years	N/A	14 years (i.e., excluding 3 years of discontinuation)	2 years	20 years	8 + 2 years
**HU starting age**	26 years	N/A	29 years	25 years (?)	20 years	34 years
**Age at MDS diagnosis**	41 years	55 years	47 years	34 years	40 years	45 years
**Time since HU start (time since HSCT)**	15 years	N/A	18 years	9 years (7 years)	20 years	11 years (3 years)
**2016 WHO classification**	MDS-EB2/AML (authors RAEB-2/AML)	MDS-EB1 (authors: AML with myelodysplasia-related changes)	MDS-MLD/AML (authors: MDS/erythroid leukemia	MDS-MLD	MDS-EB1	MDS-EB-2
**Presentation**	Refractory pancytopenia	anaemia and erythroblastosis	Severe macrocytic nonregenerative anemia	Progressive anaemia, thrombocyto-penia	Progressive anaemia, thrombocyto-penia	Anemia, neutropenia
**Peripheral blood smear**	Not detailed	Agranular blast cells with high nucleocytoplasmic ratio, hypochromic erythrocytes, some with Pappenheimer bodies, ring sideroblasts	Holly-Jowel bodies, poikilocytosis, no blast	Red blood cell distortion, nucleated red blood cells, no blast	Holly-Jowel highly hypochromic erythrocytes, neutropenia, NRBC, no blasts	3–9% blast-like cells
**Bone marrow examination**	15% myeloblasts (15% non-erythroid non- lymphoid cells)	hypercellularity (66%), 65% erythroblasts, 8 % myeloblasts (55% non-erythroid non-lymphoid cells), dysgranulopoiesis	no excess blasts, dysmyelopoiesis in 3 cell lineages	hypercellularity (95%), 2% blasts, erythroid and megakaryocytic dysplasia, erythroid hyperplasia and left-shifted myelopoiesis	Lipocytes almost absent. All three lineages represented with hyperplasia and dyserythropo-iesis of the red blood cell lineage (granulocytes/RBC ¼), maturation of the granulocytes with left shift and dysmegakaryopoiesis, with elevated number of megakaryocytes. 6% blasts of the granulocyte lineage (CD34 stain). Intermediate bone marrow infiltration, 12–15%, by T lymphocytes CD3, CD2, CD8, CD5, CD56 (partially)- positive with CD7 loss and CD57, TdT, TIA-1 negative. A few small B lymphocytes	10% malignant myeloblasts
**Chromosomal** **analysis**	42XY with complex cytogenetics including t(5:18), del(7)(q21) and monosomy 17	Complex abnormality with monosomy 5 and 7, del(17 p)	Monosomy 20, abnormalities 5 q, 17 p, 17 q	45,XY,-2, der(7)(2pter- > 2p11.2::7p11.1- >7q22::?2q11.2- > 2qter),inv(9)(p11q13)c [18]/45, idem, ?del(20)(q11.2q13.1)[	25 metaphases were analysed: 1 metaphase had karyotype 46,XY 24 metaphases had karyotype 42,XY,-3,der(5)t(3;5)(q21;q15), -7, der(12)t(7;12)(q11.2;p11.2~13), -16,-17, -8,add(18)(p11.3),+?21[24]/46,xx [1]	Monosomy 7 and structurally abnormal chromosome 19p (pre-conditioning BM negative for monosomy 7 and mutations associated with myeloid disorders
**Transformation to AML**	Transformation to AML after 34 days	Not reported	Transformation to AML after 2 years	none	no transformation	Transformation to AML during initial MDS treatment
**Treatment**	Induction chemotherapy with idarubicin and cytarabine, then high cytarabine in anticipation of bone-marrow transplant-ation	Not reported	Induction chemotherapy with cytosine arabinoside and etoposide	Conditioning regimen including busulfan, fludarabine, and HSCT graft	corticosteroids and ciclosporin	5-Azacytadine and decitabine, then after AML diagnosis, induction chemotherapy of idarubicin/cytarabine, followed by reinduction with cladribine, high-dose cytarabine, and granulocyte colony-stimulating factor, then myeloablative doses of melphalan, fludarabine, and total-body irradiation, followed by an HLA-haploidentical HSCT and cyclophosphamide posttransplant.
**Outcome**	Pancyto-penia, sepsis, subarach-noidal haemorrhage, death	Not reported	Disease progression with CNS involvement, death	Alive after 21 months	Death 3 months after diagnosis	Relapse 6 months after HSCT

^&^ AML: Acute myeloid leukemia, Hb: hemoglobin, HSCT: hematopoietic stem-cell transplantation, HU: hydroxyurea, MDS: myelodysplastic syndrome, MDS-EB1: myelodysplastic syndrome with excess blast type 1, MDS-EB2/AML: myelodysplastic syndrome with excess blasts type 2/acute myeloid leukemia, MDS/ML: myelodysplastic syndrome with multilineage dysplasia, N/A: not applicable.

## Data Availability

Not applicable.

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
