# Peer review of "Case Report of Myelodysplastic Syndrome in a Sickle-Cell Disease Patient Treated with Hydroxyurea and Literature Review"

_biomedicines, 2022, doi:10.3390/biomedicines10123201_

Round 1

Reviewer 1 Report

This manuscript report one case of myelodysplastic syndrome in sickle cell disease after HU treatment, and compare this case with others 5 cases to obtain further insights into the pathogenesis of the preleukemic process without the involvement of additional abnormalities acquired during the transformation into AML. However, the authors only summarized some male cases, which is not convincing enough to support the author's conclusions.

 1.     The cases mentioned in the manuscript are too few to support the authors' conclusions. If possible, please add more cases.

2.     As mentioned by the authors, the cases mentioned in the manuscript are all male. Whether the authors' conclusions are only applicable to male, the authors had better include some female cases to better illustrate the conclusions.

3. Table 1 looks weird, the authors are recommended to adjust the table to one page, which is easy to read.

4.     It is recommended that the author replace Figure 1 with a high-quality image.

Author Response

We hereby respectfully submit our revised version of our case report manuscript entitled “Case report of myelodysplastic syndrome in a sickle-cell disease patient treated with hydroxyurea and literature review” in Biomedicines as a research article. Our paper was strictly revised according to the reviewer’s comments and we thoroughly read our manuscript to correct some minor typos and improve English language.

Reviewer 1

  • The cases mentioned in the manuscript are too few to support the author’s conclusion. If possible, please add more cases.

We first would like to thank the reviewer 1 for his relevant comments and suggestions that would undoubtedly help in enhancing clarity of our manuscript.

This submitted work is a case report of myelodysplastic syndrome in a 40-year-old male SCD patient treated with hydroxyurea (hydroxycarbamide, HU). Importantly, we made a comparison of this single case coming from the ESCORT-HU study with previously published similar cases that we found in the current existing literature (referenced in details in Table 1). To the best of our knowledges, there is no existing additional data explaining why: 1) we could not add more cases in our manuscript, and 2) we specifically wrote in the abstract part line 25-26 that: “did not allow to draw any significant conclusions due to the paucity of these events. Given these rare described events, this last sentence allows us not to draw any substantial conclusion and clearly highlights the fact that some additional studies with a longer follow-up are still in need as indicated line 191 in the conclusion part: “… but additional surveillance with a longer follow-up is required”.

  • As mentioned by the authors, the cases mentioned in the manuscript are all male. Whether the authors’ conclusion are only applicable to male, the authors had better include some female cases to better illustrate the conclusions.

We would like to draw the reviewer's attention to the fact we made an exhaustive and thorough review of the current existing literature dealing with this theme and to the best of our knowledges, we did not find any other existing published study. In addition, in no way we had the intention to deliberately include only male cases in our comparative review as displayed in Table 1, but rather these are the only one published we found. So, we cannot include female cases as asked since these data are currently unavailable.

  • Table 1 looks weird, the authors are recommended to adjust the table to one page, which is easy to read.

We thank the reviewer 1 for his suggestion to enhance clarity of our submitted work. However, we just followed MDPI’s guidelines and template to create the displayed table 1.

  • It is recommended that the authors replace Figure 1 with a high-quality image.

We thank the reviewer 1 for his suggestion and have replaced Figure 1.

Reviewer 2 Report

This is case report of patients with MDS in the course of HU treatment his SCD.

The case was out of many described cases of SCD and risk of myeloid neoplasms was discussed. It remind us that such complication can happen so I don't have any remarks.

Author Response

We hereby respectfully submit our revised version of our case report manuscript entitled “Case report of myelodysplastic syndrome in a sickle-cell disease patient treated with hydroxyurea and literature review” in Biomedicines as a research article. Our paper was strictly revised according to the reviewer’s comments and we thoroughly read our manuscript to correct some minor typos and improve English language.

Reviewer 2

This is case report of patients with MDS in the course of HU treatment his SCD. The case was out of many described cases of SCD and risk of myeloid neoplasms was discussed. It remind us that such complication can happen so I don’t have any remarks. 

We would like to thank the reviewer 2 for his relevant comment.

Round 2

Reviewer 1 Report

I think the manuscript has been sufficiently improved to warrant publication in Biomedicines.